# Strengthening Polylactic Acid by Salification: Surface Characterization Study

**DOI:** 10.3390/polym15030492

**Published:** 2023-01-17

**Authors:** Jessica Schlosser, Michael Keller, Kamran Fouladi, Babak Eslami

**Affiliations:** Department of Mechanical Engineering, Widener University, Chester, PA 19013, USA

**Keywords:** surface characterization, scanning probe microscopy, additive manufacturing, PLA, multifrequency AFM

## Abstract

Polylactic acid (PLA) is one of the market’s most commonly used biodegradable polymers, with diverse applications in additive manufacturing, specifically fused deposition modeling (FDM) 3D printing. The use of PLA in complex and sophisticated FDM applications is continually growing. However, the increased range of applications requires a better understanding of the material properties of this polymer. For example, recent studies have shown that PLA has the potential to be used in artificial heart valves. Still, the durability and longevity of this material in such a harsh environment are unknown, as heart valve failures have been attributed to salification. Additionally, there is a gap in the field for in situ material characterization of PLA surfaces during stiffening. The present study aims to benchmark different dynamic atomic force microscopy (AFM) techniques available to study the salification phenomenon of PLA at micro-scales using different PLA thin films with various salt concentrations (i.e., 10%, 15%, and 20% of sodium chloride (NaCl)). The measurements are conducted by tapping mode AFM, bimodal AFM, the force spectroscopy technique, and energy quantity analysis. These measurements showed a stiffening phenomenon occurring as the salt solution is increased, but the change was not equally sensitive to material property differences. Tapping mode AFM provided accurate topographical information, while the associated phase images were not considered reliable. On the other hand, bimodal AFM was shown to be capable of providing the topographical information and material compositional mapping through the higher eigenmode’s phase channel. The dissipated power energy quantities indicated that how the polymers become less dissipative as salt concentration increases can be measured. Lastly, it was shown that force spectroscopy is the most sensitive technique in detecting the differences in properties. The comparison of these techniques can provide a helpful guideline for studying the material properties of PLA polymers at micro- and nano-scales that can prove beneficial in various fields.

## 1. Introduction

The idea of artificial heart valves made of polylactic acid (PLA) is getting closer to reality. However, the extent of degradation of plastic valves by salification is not well understood and has not been extensively investigated. The present study is focused on investigating the impact of salification on the valves through a comprehensive material characterization effort utilizing various AFM methods.

### 1.1. Atomic Force Microscopy Techniques

Atomic force microscopy (AFM), invented in the 1980s, is from the family of scanning probe microscopies providing major advantages in material characterization. AFM is capable of extracting topographical information from a wide range of samples (conductive to nonconductive, metals to polymers) while allowing material property characterization to be performed, such as mechanical properties, chemical properties, and electrical and magnetic properties [1,2,3,4,5]. The main component of AFM is the micro-cantilever. The deflection of the cantilever is measured by shining a laser over the back of the cantilever and measuring the signal through the laser’s reflection on the photodiode detector. The analysis and manipulation of this signal will differ based on the imaging scheme. Figure 1 depicts a schematic of how dynamic AFM imaging works. One specific mode of AFM is the tapping mode, in which the cantilever oscillates at the cantilever’s resonance frequency (i.e., the first eigenmode frequency) using a piezoelectric driver. Although the excitation frequency is fixed, the distance between the tip and sample is modulated so that the oscillation amplitude remains constant. In this method, named amplitude modulation AFM (AM-AFM), the height (i.e., topography) information is gathered from the amplitude signal [6]. The phase signal, the phase lag between excitation at the base of the cantilever and the oscillation of the tip, provides information on the compositional mapping of the surface. Tapping mode can be carried out in two different regimes, namely, attractive or repulsive. Attractive mode is when the phase signal is greater than 90°, and repulsive mode is when the phase signal is less than 90° [7]. In the attractive regime, the cantilever does not physically touch the surface but interacts with long-range interaction forces such as electrostatic forces or van der Waals forces. The *bistability* phenomenon will occur if the oscillation setpoint and free oscillation amplitude are set so that the solution of the equation of motion of the cantilever can have two roots [8]. Ideally, the repulsive regime creates higher quality images, but the attractive regime can be good for soft samples, as it applies smaller forces on the sample.

Although tapping mode AFM can provide both topography and compositional mapping, there is no guarantee that the quality of height and phase channels will be good for all samples. Additionally, the information observed from phase images is not directly related to mechanical properties, rendering them not quantitatively useful. Therefore, there has been significant enthusiasm toward force spectroscopy and material models needed to fit the data during the past few years. In force spectroscopy techniques, the cantilever is approached to the sample, while its deflection is measured, as shown in Figure 1 [9,10,11], going through the attractive and then repulsive forces. Using the feedback loop, the cantilever is pulled out of the sample after reaching the given maximum deflection. The observed deflection can then be converted into force versus tip sample separation to extract mechanical properties such as adhesion, stiffness (modulus), and the indentation depth. Force spectroscopy only collects a single point measurement, so force mapping is useful to collect a grid of force curves over an area of a sample and allows for the mechanical properties within that area to be compared [5,12].

In the early 2000s, it was found AFM tapping mode might not provide both topography and material composition at the same time when imaging the soft matter [13,14]. Therefore, a branch of imaging techniques known as multifrequency AFM was introduced [15]. Bimodal AFM, the most common multifrequency technique, is accomplished through the simultaneous excitation of two eigenmode frequencies of the AFM cantilever. This technique simultaneously generates multiple additional properties, such as a second phase channel. It can also allow for higher resolution and more detailed information about the sample. Subsequently, it was shown that higher eigenmode AFMs could provide depth penetration through samples and enhance AFM capabilities to image subsurface features. 

Many AFM techniques have been developed and used over the years. However, there is still a need to fundamentally understand the strengths and limitations of these techniques. This work aims to fill this knowledge gap when dealing with biodegradable polymers.

### 1.2. 3D-Printed Artificial Heart Valves

Cardiovascular disease (CVD) is the current leading cause of death in the United States, resulting in 696,962 deaths in 2020 [16]. Heart valve disease (HVD), which concerns any of the four heart valves, is a major contributor to CVD. Moreover, the aortic valve can thicken and become stiff (aortic valve stenosis), which is the most common cause of the need for surgery. The stiffening causes the valve to be unable to open fully, reducing the blood flow to the body [17]. Aortic valve stenosis is commonly caused by calcium and calcium phosphate buildup on the aortic valve, otherwise known as salification. Therefore, it is critical to firstly understand the stiffening phenomenon on the polymer surface of an artificial aortic valve and secondly visualize this concept to be able to design better valves. 

Generally, two types of heart valves can be used for aortic valve replacement—biological or mechanical. Biological or bioprosthetic valves replace the original aortic valve with a new valve typically made from bovine (cow) or porcine (pig) tissue [17]. This animal tissue is referred to as a xenograft. Similarly, allografts are valves taken from human cadavers or brain-dead patients [17]. However, these valves are less readily available as they cannot be mass-produced or available commercially. 

Mechanical valves, generally made of carbon or other sturdy materials, last longer than biological valves and do not need to be replaced. However, they do require the patients to take blood-thinning medications for the rest of their lives. Recent emerging technologies have turned aortic heart valve replacement towards 3D printing. It is believed that the interaction between the artificial heart valve and the patient’s native anatomy could be significantly improved through the 3D printing of patient-specific models. Unlike mechanical or biological heart valve replacements, 3D-printed heart valves can be designed specifically for the size and anatomy of the patient. Although salification is a concern for all heart valve types, it can play a major role in the longevity and performance of 3D-printed valves. However, there is little understanding of if and how the salification process stiffens plastic valves.

### 1.3. Motivation and Objectives

Recent studies in the cardiovascular field are touting artificial heart valves made of polylactic acid (PLA), a biodegradable polymer commonly used in additive manufacturing and extensively in fused deposition modeling. However, 3D-printed heart valves can be negatively impacted by salification, similar to other artificial heart valves. Moreover, the extent of degradation of plastic valves by salification is not well understood and has not been extensively investigated. 

In the present study, we undertake a comprehensive surface characterization effort focused on the salification process of biodegradable polymer-based heart valves. More specifically, our research is focused on how salification affects PLA thin films at the microscopic level. Various AFM methods are utilized to help gain a better understanding of how salification affects the material properties and quality of PLA at both quantitative and qualitative levels. The methods include tapping mode images, multifrequency AFM, energy-based AFM analysis, and force spectroscopy. The range of AFM methods in the present study can be used to provide a comprehensive guide to researchers studying salification effects on PLA to generate the most meaningful information using AFM.

The knowledge gained by the present investigation will help to understand if 3D-printed valves are viable for heart valve replacement. The findings can also help determine if salification can strengthen PLA for additive manufacturing applications. If results show that salification has caused stiffening on the surface of the PLA polymer, one can decide how to treat the polymer to reduce the chance of salt bonding to the surface.

## 2. Methods and Materials

Three PLA film films with increasing salt concentrations were produced to understand how salification affects the material properties of PLA. Although there is a difference between bulk and surface material properties, it is understood that the performance of artificial heart valves is dictated by the changes on their surfaces. Additionally, for experimental purposes, spin-coated thin films provide more controlled methods of measurement for fundamental understanding of the matter. Initially, small beads of PLA were placed in a glass beaker, and methylene chloride was added to dissolve the polymer. A magnetic stirrer was then added, and the sample was mixed until all the PLA was dissolved. Three sodium chloride (NaCl) solutions were then prepared. The first solution had a salt to di-water weight ratio of 1:10, the second solution had a ratio of 1.5:10, and the third solution had a ratio of 2:10. With a pipet, equal parts of sodium chloride solution and PLA-methylene chloride were combined in a beaker. This procedure was repeated for each sodium chloride solution. Dime-sized silicon wafers were then prepared through a cleaning process with an ultra-sonic cleaner and an isopropyl and de-ionized water rinse. Each silicon (Si) wafer was then placed on the SCK-300 digital spin-coater device. The PLA-methylene chloride-salt solution was deposited on the Si wafers. The sample was then spun at 3000 rpm for 60 s. This process was repeated for each of the three NaCl concentrations. Each sample was placed on a glass slide and set aside overnight to fully dry. The samples are referenced in Table 1. 

Once the samples were ready for measurement, the Asylum Research MFP-3D AFM controlled with an ARC2 controller was used to perform various AFM studies. The same type of cantilever (Multi75-G), manufactured by Budget Sensors, was used throughout the measurements for proper comparison of the AFM results. The cantilever’s spring constant was measured by Sader’s method [18] and was found to be 2.07 N/m with a resonance frequency of 78 kHz. These cantilevers are around 225 μm in length, 28 μm in width, and 3 μm in thickness. The tip radious is around 10 nm and made out of silicon.

The conventional tapping mode was the first method used to measure each sample’s topography and phase. Simultaneously, the observables of the conventional tapping mode were analyzed from the energy-based method. Specifically, the amplitude and phase signals collected from tapping mode imaging were converted into virial (*V_ts_*) and dissipated power (*P_ts_*), which are convolutions of the tip–sample interactions with position and velocity, respectively. Equations (1) and (2) describe the conversions, where index *i* specifies the corresponding eigenmode under study, k is the stiffness, A the instantaneous amplitude, $A_{0}$ the free amplitude, f_exc  the excitation frequency, $f_{0}$ the free resonance frequency, ϕ the phase, and Q the quality factor. Based on these equations, if the imaging mode is simple tapping mode where the excitation frequency is equal to the resonance frequency, the $f_{exc}/f\;$ will be reduced to one and will simplify the equations [19,20].
(1)$$V_{ts,i}=\frac{k_{i}A_{i}}{2}\left[A_{i}\left(1-\frac{f_{exc,i}^{2}}{f_{i}^{2}}\right)-\frac{A_{0,i}}{Q_{i}}cos\phi_{i}\right]$$
(2)$$P_{ts,i}=\frac{\pi f_{exc,i}k_{i}A_{i}^{2}}{Q_{i}}\left[\frac{A_{0,i}}{A_{i}}sin\phi_{i}-\frac{f_{exc,i}}{f_{i}}\right]$$

The second AFM method performed on each of the samples was a multifrequency technique called bimodal AFM. In this method, imaging is performed through the simultaneous excitation of two different eigenmodes, while the first eigenmode is modulated with the feedback loop, and the second eigenmode is in an open loop. The technique results in a series of images, including topography, the first eigenmode amplitude, the second eigenmode amplitude, the first eigenmode phase, and the second eigenmode phase. Since one can optimize the two relatively weakly coupled eigenmodes separately, both topography and compositional mapping are guaranteed through this technique. In this method, the free amplitude of the higher eigenmode is the key parameter governing the intensity of the tip–sample interactions for a given free amplitude and amplitude setpoint of the fundamental eigenmode [21]. The user can also control the drive frequency, which should be selected close to the eigenmode frequencies. By observing the governing equation of motion of the cantilever, it can be seen that the greater indentation capability of higher eigenmodes is possible with the use of higher eigenmodes: (3)$$\frac{d^{2}\underline{z}}{d{\underline{t}}^{2}}=-\underline{z}+\frac{1}{Q}\left[-\frac{d\underline{z}}{d\underline{t}}+\cos\left(\underline{t}\right)\right]+\frac{F_{ts}\left({\underline{z}}_{ts}\right)}{kA_{0}}$$
where $A_{0}$ is the free amplitude, $\underline{z}=z/A_{0}$ is the normalized tip displacement, ${\underline{z}}_{ts}$ is the normalized tip–sample distance, $\underline{t}=\omega_{0}t$ is the dimensionless time, k is the cantilever force constant, and $F_{ts}$ is the tip–sample force interaction. The free oscillation amplitude is assumed to be equal to $\frac{F_{0}Q}{k}$ based on the work done by Ricardo Garcia et al. [21]. The damping and excitation terms are combined with the $\frac{1}{Q}$ factor. As shown in Equation (1), the last term on the right-hand side is normalized by the product of the cantilever force constant and the free oscillation amplitude. Therefore, as the denominator of this fraction increases, the effect of tip–sample force interactions on the dynamics of the cantilever diminishes. For the bimodal AFM case, the force constant of higher eigenmodes is approximately 36 times the first eigenmode. Therefore, the cantilever becomes less sensitive to forces when excited with two eigenmodes. Consequently, surface penetration is observed, and if the soft matter under study can be compressed, the AFM tip will compress the film and a stiffer surface is observed [22].

The method of force spectroscopy was carried out to understand how increased salification affects the stiffness of PLA. An approach curve was captured on a stiff glass slide while the deflection in volts was measured. To collect the appropriate data, an area of 5×5 μm was imaged using tapping mode AFM, which was also for virial and dissipated power analyses. Next, we zoomed into the image twice on both a PLA area and salt contamination area and collected a smaller image around 1×1 μm. Once we ensured that the image focused on one material in particular, a force map of 20 contact mode force curves was collected over an area of approximately 200×200 nm. The same process was repeated twice per three samples. 

Contact mechanics models were used to extract material properties of the samples by fitting the force curves to the appropriate model. The three primary models included Hertz, Derjaguin–Muller–Toporove (DMT), and Johnson–Kendall–Roberts (JKR) [23]. The Hertz model neglects adhesion and friction, which works well for nanoindentation and liquid imaging; however, it does not work for most AFM tip–sample interactions, since the AFM tip does interact with the sample. Neutral atoms and molecules still experience forces between one another, so adhesion cannot be neglected. The DMT model is appropriate for stiffer, longer-range adhesion, while the JKR model is appropriate for stronger and shorter-range adhesion. Based on this information, the DMT contact model was selected for force spectroscopy analysis.

## 3. Results and Analysis

AFM tapping mode was the first method utilized to collect information from each of the three calcified PLA samples. Figure 2 compares the results of this method for both topography and phase. Although the topography images show some increases in surface roughness as we increased the salt concentration on the samples, they are not completely distinguishable. The main differences between the results were seen in the phase results. It should be noted that when the cantilever was tuned in air (not interacting with the surface), the phase value near or at the resonance frequency was about 90 degrees. Based on the equation of motion of a simple harmonic excited system, as the cantilever interacted with the surface (stiffer than air), the phase value decreased to values below 90 degrees. Therefore, a general rule in analyzing phase images states that the lower phase values (i.e., darker colors in phase images) represent stiffer surfaces. There were clear regions where the salt contamination on surfaces was shown. This was shown as islands on samples 2 and 3. However, more importantly, an overall increase in the stiffness of samples was shown going from sample 1 to sample 3. Overall lower phase values among samples represented the stiffer surface and tip–sample force interactions, while the contrast between the islands represented the material composition.

For sample 1, which had a 10% salt concentration, the salt presented itself in small, scattered circles over 6.74% of the surface. The size and shape of these salt particles appeared to be very similar to the size and shape of the humidity pores depicted by the dark circles in the topography. Even where those pores presented themselves in the topography, the phase images showed that they were the same material as where the raised surfaces occurred. Therefore, the conclusion can be drawn that at the microscale, the salt in a calcified PLA sample with a 10% salt concentration binds to both the surface of the PLA as well as the pores.

For sample 2, which had a salt concentration of 15%, the salt presented itself in large, scattered islands over 25.44% of the surface. Compared to samples 1 and 3, the salt concentrations were distributed more evenly over the surface of the PLA. Here, the salt adhered to the actual surface of the PLA rather than distributed into the pores or binding with itself.

For sample 3, which had a salt concentration of 20%, the salt presented itself in one large island with a few scattered and raised circular patches over 33.44% of the surface. The separation of the PLA and salt concentrations was likely caused by an over-concentration of the salt solution with the PLA. The solutions were likely separated even before the spin-coating process occurred, causing the salt to bind to itself and therefore adhere to the surface of the PLA as singular large islands rather than small scattered islands, as seen in sample 2.

The subsequent analysis that was performed was a virial and dissipative analysis based on the amplitude modulation channels from the tapping mode images collected from each sample. Figure 3 displays the results of this study. 

By visual inspection, both the virial and dissipated power images yielded similar contrasts to the tapping mode phase images. The dissipated power analysis nearly depicted the same images, while the virial depicted slightly more detail [24,25]. Again, visually, it was difficult to interpret a change in stiffness from these images, but further numerical analysis could show that the results correlated with an increase in stiffness across samples. 

The next method used to analyze the calcified PLA samples was bimodal imaging, where the results were generated through the simultaneous excitation of two eigenmodes of the AFM cantilever. Figure 4 displays the results of this method, including the topography, phase 1 generated through the first eigenmode, and phase 2 generated through the second eigenmode. 

The bimodal imaging results shown in Figure 4 compared well with the results from normal tapping mode presented in Figure 2, and the topographies appeared to be nearly identical. Similarly, there was some increase in surface roughness as the salt concentration increased, but the materials were not completely distinguishable. The phase 1 images were also nearly identical to the phase images from Figure 2, which clearly distinguished the salt contaminations from the PLA surface. Moreover, areas of the surface became more detailed when considering the phase 2 images based on the second eigenmode. This change was most notable for samples 2 and 3, as the gaps in the salt contaminations began to become more apparent. 

Additionally, the raised area in the top right corner of sample 3 differed between phase 1 and phase 2. The phase 1 image showed that area as PLA or an area with lower stiffness, while salt contaminations began to appear in that region on the phase 2 image. Overall, comparing these sets of images, bimodal imaging allows for a more detailed understanding of the materials present on the surface compared to normal tapping mode. 

The final analysis performed on the three calcified PLA samples was force spectroscopy, which allowed for the Young’s modulus, or stiffness, of each sample to be collected. A series of force curves was collected using force mapping over an area of high salt concentration and primarily PLA. For example, for sample 3, force maps were collected in the dark purple outer region and the pink inner region depicted in Figure 5a. Figure 5b is an example of two force versus separation curves for the different areas on sample 2, which were converted from the raw deflection versus distance curves collected through force spectroscopy. Figure 5c displays the effective Young’s modulus values for each sample, which were calculated by fitting the DMT contact model to every curve and performing a particle analysis to determine the percentage of the surface covered in salt. Figure 5d is a physical representation of the DMT contact model, in which the DMT spring is represented as an infinite series of springs. Based on this representation, more springs are activated as the AFM tip goes deeper into the surface. Therefore, stiffness depends both on the position of the tip and the contact area. 

The 3D representation of the topography superimposed with the phase clearly distinguished the areas of high salt concentrations as well as the impact on the roughness of the samples. Additionally, the slopes of the force versus separation curves over those two different areas distinguished the stiffnesses. The orange-colored force curve over the salt contamination area was steeper than the purple curve over the PLA area. This trend was consistent over each sample, and the Young’s modulus over the PLA areas also increased from sample 1 to sample 3. Therefore, as the salt contaminations covered a greater percentage of the surface and the overall NaCl concentration increased, the effective Young’s modulus of the samples also increased. 

It is important to note the quantitative and qualitative differences when analyzing and comparing the results from each method, including tapping mode, a virial and dissipative analysis, bimodal imaging, and force spectroscopy. First, in examining the phase change across the samples, specifically the PLA regions, it could be interpreted that the stiffness of the PLA decreases as the NaCl concentration increases. For example, as shown in Figure 5a, the phases for the PLA regions of samples 1, 2, and 3 were approximately 35°, 30°, and 15°, respectively. However, the force spectroscopy results proved otherwise. The effective Young’s modulus was calculated as an accumulation of force spectroscopy results for both PLA and salt contamination regions on each sample along with their respective percent surface area. Through this analysis, the force spectroscopy results gathered that the average Young’s modulus over the PLA regions increased across the samples as the NaCl concentration increased. Specifically, the average Young’s modulus in these regions for samples 1, 2, and 3 were 0.774 GPa, 1.062 GPa, and 1.154 GPa, respectively. The standard deviations were 0.53 kPa, 0.79 kPa, and 0.61 kPa, respectively. A similar trend could be seen for the salt contamination areas, where phase images indicated that the pink salt areas of sample 3 had a lower stiffness than the yellow salt areas of samples 1 and 2. Once again, the force spectroscopy results proved otherwise, as sample 3 had the highest average Young’s modulus value of 2.582 compared to values for samples 1 and 2 of 1.718 GPa and 1.373 GPa, respectively. It is also worth noting that these phase trends from tapping mode were consistent with the phase 1 and phase 2 images from bimodal imaging. 

Theoretically, each AFM method should provide the same information about each sample. However, the results proved there were differences. While there were similarities between the methods, each method offered additional unique information and could provide guidelines for researchers when choosing different characterization techniques for samples with varying mechanical properties. Figure 6 displays the data normalization results comparing the results of each AFM method to understand the sensitivity of each AFM characterization technique performed. Sample 2 and sample 3 data were normalized by sample 1 data for the corresponding method of measurement. For example, in the tapping mode AFM column, sample 2 and sample 3 average phase values are divided by sample 1 average phase values. Since sample 1 is the untreated polymer, we used this sample as the reference point in our study. The closer the value was to one in this plot, the less of a difference was observed by the measurement technique. The dotted horizontal line represents the threshold. The average phase for simple tapping mode, average dissipated power, average phase 2 for bimodal imaging, and the effective Young’s modulus from force spectroscopy were selected for comparison. 

Overall, the sensitivity study shows that the phase results from simple tapping mode are not reliable. Although they provide a good sample topography, the phase images are not necessarily reliable regarding the stiffness of the samples. The energy quantity results do provide useful information as the sample surfaces become less dissipative, indicating increased stiffness as salt concentration increases. Additionally, the bimodal phase 2 images do provide useful information about the samples. The results show that the higher the salt concentration, the lower the phase values, indicating that stiffness increases across the samples. Finally, the force spectroscopy results follow the trend exactly as the Young’s modulus and stiffness increase across the samples. These results also show that the force spectroscopy results are more sensitive to the sample change.

Conducting each method proved that using the same technique across samples of increasing stiffness may not be viable. It also shows that some techniques, such as bimodal phase imaging and force spectroscopy, provide more useful information than others, such as simple tapping mode phase. In other words, force spectroscopy and bimodal AFM are more sensitive to material differences over a given surface. Finally, the comparison between the normal tapping mode images and bimodal imaging proves that both methods do not need to be conducted. It would be more useful for the researcher to only consider bimodal imaging, as the addition of phase 2 and the implementation of two eigenmodes provide similar yet more accurate information about the samples. This holds true as long as the product of $kA_{0}$ explained in Equation (1) does not increase drastically, so the forces applied on soft matter cause damage to the surface.

## 4. Conclusions

In this study, we compared tapping mode AFM, bimodal AFM, energy analysis (dissipated power), and force spectroscopy techniques while characterizing one of the most commonly used biodegradable polymers (PLA). During this study, it was shown that as the salt concentration on PLA surfaces increases, AFM techniques are capable of detecting the material property differences. However, it was also shown that each technique has its own sensitivity to these property changes. Tapping mode AFM is not a reliable characterization technique for material properties. However, using the amplitude and phase signals of tapping mode AFM, we derived the virial and dissipated power, which verified that dissipated power (a combination of amplitude and phase information) is more sensitive to sample differences. In addition to simple tapping mode AFM, bimodal AFM was shown to be a useful technique that can detect the material changes while still providing topographical information. However, its sensitivity is not as good as force spectroscopy analysis. This study concluded that in order to detect different material properties, force spectroscopy is the most sensitive technique, although it cannot provide topographical information. Therefore, based on this work, it is recommended that investigators perform bimodal AFM imaging, followed by a force map, that can fit different material models discussed in the paper for a comprehensive analysis. 

## Figures and Tables

**Figure 1 polymers-15-00492-f001:**
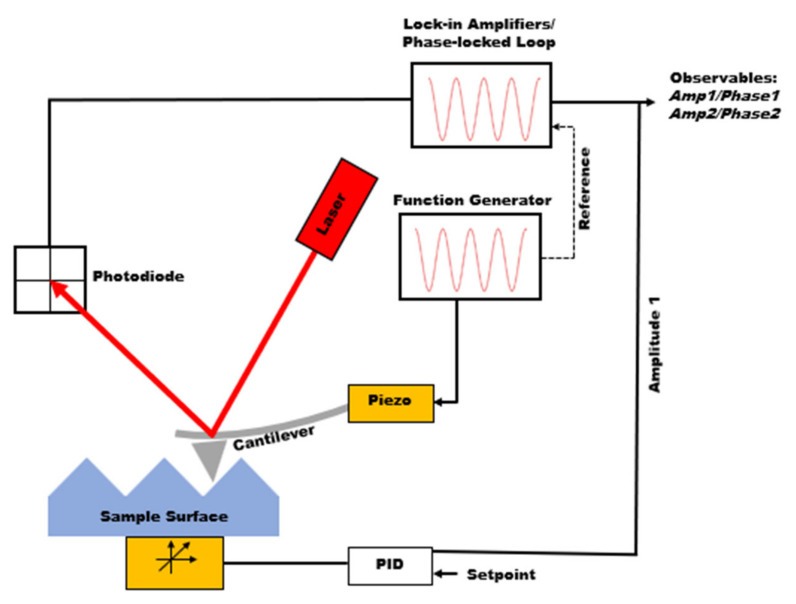
Schematic of dynamic atomic force microscopy.

**Figure 2 polymers-15-00492-f002:**
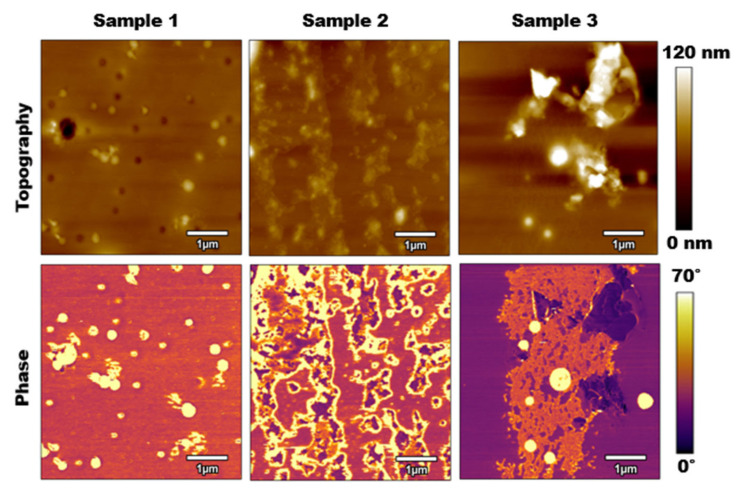
Tapping mode AFM results: **Top Row**: topography images for samples 1, 2, and 3 from left to right. **Bottom Row**: Phase images for samples 1, 2, and 3 from left to right. Scan sizes are 5 μm×5 μm with a scan rate of 1 Hz, free oscillation amplitude of 100 nm, with 60% setpoint.

**Figure 3 polymers-15-00492-f003:**
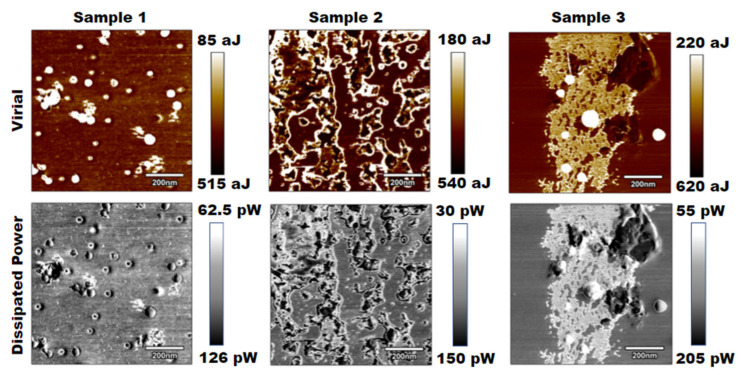
Results for virial and dissipated power for three PLA samples of increasing salt concentrations.

**Figure 4 polymers-15-00492-f004:**
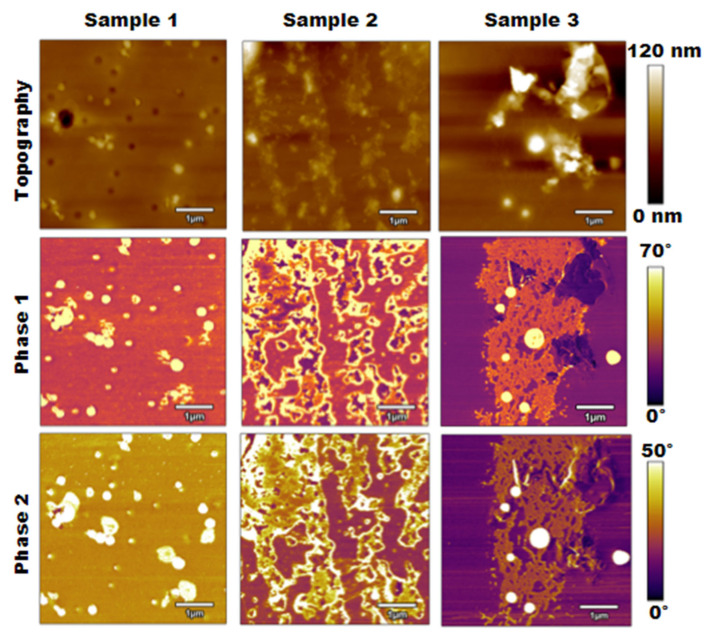
Topography, phase 1, and phase 2 results for bimodal imaging of three PLA samples of increasing salt concentration.

**Figure 5 polymers-15-00492-f005:**
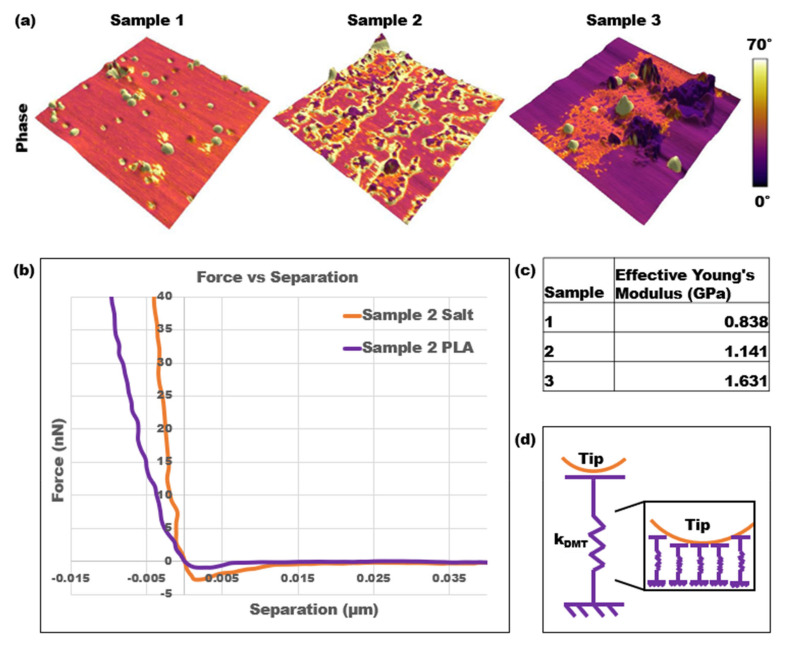
Force spectroscopy results: (**a**) three-dimensional topography of three PLA samples overlayed with a color scale representing the phases; (**b**) two force curves of sample 2 on an area of high salt concentration vs. a primarily PLA area; (**c**) effective Young’s modulus results using force spectroscopy and the DMT contact model; (**d**) spring and AFM tip representation of the DMT contact model.

**Figure 6 polymers-15-00492-f006:**
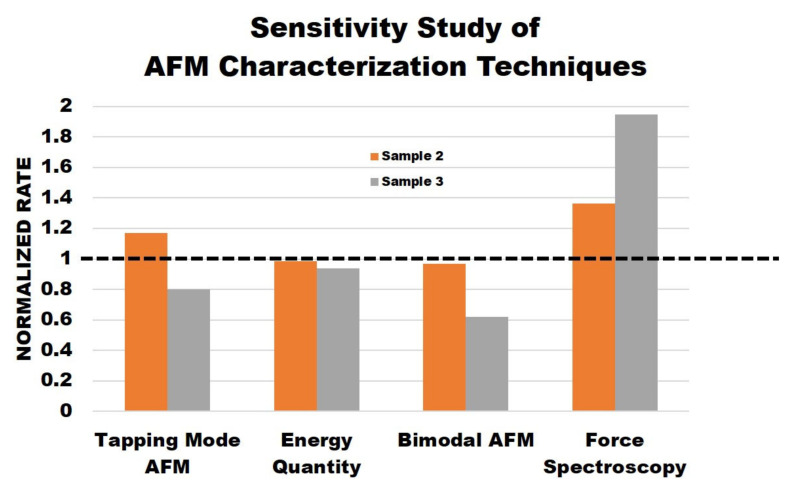
Normalized data of sensitivity study of AFM characterization techniques.

**Table 1 polymers-15-00492-t001:** Sample NaCl concentration with given sample numbers.

Sample Number	% Weight NaCl Concentration
1	10
2	15
3	20

## Data Availability

Data will be available upon request.

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
