# Peer review of "Strengthening Polylactic Acid by Salification: Surface Characterization Study"

_polymers, 2023, doi:10.3390/polym15030492_

Round 1
Reviewer 1 Report
This manuscript reports on the characterization of polylactic acid (PLA) by different AFM characterization modes. It is stated that PLA is one of the most common biodegradable polymers but that the impact of salification needs to be better investigated and understood.
The objectives seem to be clear, on the one hand the understanding of the impact of salification on PLA valves, and on the other, the characterization by AFM methods. But then, the results need to be better discussed. There is no comparison to literature or to previous results and even there is no discussion on the supposed objectives. As conclusion: what is the impact of salification on the valves?
In addition to this general important correction which needs to be implemented there are some other aspects which need to be improved and/or answered:
1. It is mentioned that the findings of this work will help to understand if 3D printed valves are viable for heart valve replacement but samples used are not 3D printed but films obtained by spin coating. So, one first question is whether this model is valid. It can be, and it can be mentioned (maybe) that as a model is easier and more controllable to prepare films by spin coating, but n any case information is missing as for instance thickness and roughness of these films.
2. Since AFM is relevant in this study, some additional information as cantilever/tip dimensions (and in particular tip radius) should be given.
3. Regarding the Young’s modulus: error should be added to the averaged values.
4. Results shown in Figure 6 should be better explained. How were data normalized. In respect to what?
Author Response
We would like to thank the reviewer for valuable feedback on our manuscript. We have addressed all the comments.

Reviewer 2 Report
In this manuscript, the authors describe the capabilities of the AFM technique in the investigation of the soft materials like polymers, e.g. polylactic acid. The manuscript appears to be interesting, starting from a real problem that could be found in the use of polylactic acid for the promising application in the heart valve disease. The change of the stiffness of these valves could represent a limitation on the use of polylactic acid for the production of these valves. However, I have found the manuscript rather detailed in the section of the AFM technique, while other aspects are not considered, in particular, the determination of the chemical composition of the surface that can’t be provided by AFM, at most qualitative. How do the authors consider the agglomerates composed of the added salt? Looking at the images, moving from the topography to the phase mode, it seems hard to find any difference on the details. In figure 2, in the phase images, there are three colours (purple, orange and yellow), what do they mean?
Just out of curiosity, what is the material used for the tip?
Pag. 3 line 45 “… similar to other artificial heat valves…” most probably heat must be changed in heart
Page 4 line 9 the word film in double in the sentence “Three PLA film films….”
Author Response
We would like to thank the reviewer for the valuable response. We have addressed all the comments.

Round 2
Reviewer 1 Report
All the questions and comments have been addressed and answered and there are no additional comments.